

# Association between gestational weight gain-related attitudes, pre-pregnancy weight and prenatal health behaviours

Wafaa Taha Elgzar[1], Majed Said Alshahrani[2], Nouf Salem Alshahrani[3] and Heba Abdel-Fatah Ibrahim[1]

[1] Department of Maternity and Childhood Nursing, Nursing College, Najran University, Najran, Saudi Arabia
[2] Department of Obstetrics and Gynecology, College of Medicine, Najran University, Najran, Saudi Arabia
[3] Medical-Surgical Nursing, Najran University, Najran, Saudi Arabia

## ABSTRACT

**Background.** Prenatal health behaviors and pre-pregnancy body mass index (BMI) can influence gestational weight gain (GWG) attitudes, this specific connection has not been extensively studied among pregnant women in Saudi Arabia. This study aims to explore the association between GWG attitudes, pre-pregnancy weight, and prenatal health behaviors among Saudi pregnant women.

**Methods.** A correlational cross-sectional study was conducted in Najran, Saudi Arabia, from March to May 2024. The study included 413 pregnant women. An online self-reported questionnaire was used to collect data, which involved three main parts: participants' basic data, the Pregnancy and Weight Gain Attitude Scale (PWAS), and the Prenatal Health Behavior Scale (PHBS). Logistic regression was applied to explore the association between GWG attitudes, pre-pregnancy BMI, and prenatal health behaviours.

**Results.** Logistic regression analysis showed that pre-pregnancy BMI outside the healthy range negatively predicts positive attitudes toward excessive gestational weight gain (EGWG). Being overweight [AOR = 0.434, $P = 0.017$, 95% CI [0.219–0.861]] or obese [AOR = 0.157, $P = 0.001$, 95% CI [0052–0.469]] significantly decreased the probability of positive attitudes toward EGWG when compared with normal weight. Furthermore, practising health-promoting behaviours during pregnancy [AOR = 0.132, $P = 0.001$, 95% CI [0.050–0.483]] significantly decreased the probability of a positive attitude toward EGWG.

**Conclusion.** This study emphasized the important role of pre-pregnancy BMI and prenatal health-promoting behaviours in maintaining appropriate GWG attitudes. Therefore, healthcare providers have to play a proactive role and start early by securing a normal BMI before pregnancy and educating women about prenatal health-promoting behaviours.

## INTRODUCTION

Excessive gestational weight gain (EGWG) refers to gaining more weight during pregnancy than is recommended for a woman's specific body mass index (BMI) category. The Institute

Corresponding authors
Majed Said Alshahrani,
msalshahrane@nu.edu.sa
Heba Abdel-Fatah Ibrahim,
heaibrahim@nu.edu.sa

of Medicine (IOM) guidelines formulated an individual pattern of gestational weight gain (GWG) for each pregnant woman according to their pre-pregnancy body mass index (BMI) classification. For illustration, the optimum GWG for underweight women is 12.5–18 kg, normal-weight women 11.5–16 kg, overweight women 7–11.5 kg, and obese women 5–9 kg (*Institute of Medicine US, 2009*).

Excessive gestational weight gain is considered a main modifiable risk factor that negatively impacts pregnancy outcomes. This includes having a macrocosmic baby, cesarean delivery, gestational diabetes, and pregnancy-induced hypertension (*Hrolfsdottir et al., 2015*; *Goldstein et al., 2017*). A meta-analysis involving 39 cohorts estimated that 23.9% of pregnancy complications were associated with mothers who are overweight and obese. GWG is also linked to offspring health, such as the risk of childhood overweight and obesity (*Santos et al., 2019*).

Nevertheless, EGWG is relatively prevalent, with an estimated 40 to 60% of pregnant women experiencing EGWG greater than the recommended by IOM guidelines (*Gavard & Artal, 2014*; *Institute of Medicine US, 2009*; *Deputy et al., 2015*). The IOM identified several factors that predict GWG, including maternal height, pre-pregnancy BMI, body fat percentage, living conditions, smoking habits, diabetes status, and previous pregnancies (*Victor et al., 2024a*). Following these guidelines, it is vital to ensure a healthy pregnancy. It is important to monitor and manage weight gain appropriately to avoid underweight or overweight conditions. This ensures a healthy pregnancy and reduces the risk of complications in the newborn (*Victor et al., 2024b*).

A recent World Health Organization report found that overweight and obesity rates among Saudi Arabian adults have increased significantly, reaching 60% (*Almubark & Alqahtani, 2023*). According to the cohort study conducted by *Wahabi et al. (2016)* in Riyadh, Saudi Arabia, the prevalence of overweight and obesity in women is more than 68%, which is much higher than its counterpart in Western countries (30%) and is the highest in the world (*Huda, Brodie & Sattar, 2010*). Among the factors that have contributed to the high rate of overweight and obesity, especially among women in Saudi Arabia, is the rapid economic development and expansion of urbanization, which has led to a lifestyle that depends on a diet rich in carbohydrates and fats and a lack of physical fitness (*Moradi-Lakeh et al., 2017*). The link between urbanization and rapid economic development, sedentary lifestyle and obesity have been established by previous studies (*Almubark & Alqahtani, 2023*; *Pirgon & Aslan, 2015*) Urban expansion often leads to decreased physical activity among general population and pregnant women as it promotes car-centric transportation and reduces opportunities for walking and cycling. Additionally, the abundance of fast-food restaurants and convenience stores in urban environments makes it easier to consume calorie-dense, nutrient-poor foods and sugary drinks (*Almubark & Alqahtani, 2023*; *Pirgon & Aslan, 2015*). In addition, a multicenter cohort study conducted in Riyadh, Saudi Arabia, showed that less than one-third (31.8%) had adequate GWG, 25.9% had EGWG, and 42.3% had inadequate GWG (*Fayed et al., 2022*).

During pregnancy, the woman's body undergoes massive changes within a relatively short period. Therefore, pregnancy may be associated with high vulnerability to body image dissatisfaction (*Linde et al., 2022*). The adaptation strategies will be greatly influenced by

her GWG-related attitudes and health behaviours (*Grogan, 2017*; *Savard et al., 2021*). Although the recommended GWG is necessary to secure a healthy pregnancy, a significant increase in weight that occurs during that specific time may lead to or exacerbate already existing negative attitudes toward a woman's body image (*Desmecht & Achim, 2016*). Furthermore, studies have indicated that high GWG and BMI before pregnancy were related to negative attitudes toward body image (*Andrews, Hill & Skouteris, 2018*; *Shloim et al., 2019*; *Roomruangwong et al., 2017*).

Prenatal health behaviours and pre-pregnancy BMI can contribute to GWG attitudes, though this has not been studied during pregnancy in Saudi Arabia. BMI, previously known as the Quetelet index, is calculated by dividing their weight in kilograms by their height in meters squared. It provides a general indication of nutritional status. A BMI under 18.5 is considered underweight, while a BMI from 18.5–24.9 is considered normal weight. A BMI between 25.0 and 29.9 is classified as overweight, and a BMI of 30.0 or higher indicates obesity. Obesity is further classified into categories I (BMI 30.0–34.9), II (BMI 35.0–39.9), and III (BMI 40 or higher) (*World Health Organization, 2024*). In the general population, positive attitudes toward body image have been associated with healthy lifestyles and eating behaviours (*Tylka & Kroon Van Diest, 2013*; *Jáuregui-Lobera et al., 2014*). In pregnancy, a cross-sectional study showed that pregnant women with positive attitudes toward appropriate GWG reported healthy eating habits in the third trimester (*Savard et al., 2021*). Given the health risks associated with body image issues such as EGWG, especially among women with a higher pre-pregnancy BMI, it is essential to explore pregnant women's attitudes toward GWG and how they relate to all components of prenatal health behaviours, not just nutritional behaviours. The aspects of prenatal health behaviours include health-promoting behaviours such as adequate physical activity, sleep, and consumption of healthy nutrition, as well as health-impairing behaviours such as smoking, drug use, and malnutrition during pregnancy. Therefore, this study aimed to examine the relationship between attitudes regarding EGWG, pre-pregnancy weight, and prenatal health behaviours among Saudi pregnant women. If statistical data confirms such a relationship, it will direct healthcare providers toward proactive strategies and start early by securing a normal BMI before pregnancy and educating women about health-promoting behaviours.

## MATERIALS & METHODS

### Study design
This study is designed as a correlational cross-sectional research to investigate the relationship between EGWG-related attitudes, pre-pregnancy BMI and prenatal health behaviours.

### Study setting
The study was conducted in the Najran region, which is located in southwest Saudi Arabia.

### Study participants
The study consisted of 413 pregnant women using a convenience sampling technique to recruit pregnant Saudi women in the Najran community who agreed to participate and met

the inclusion criteria. Women who had singleton pregnancies and were in the second and third trimesters of pregnancy were included in the study, as the majority of GWG occurs during the second and third trimesters (*Kominiarek & Peaceman, 2017*). However, the exclusion criteria were women having pregnancy-related complications such as gestational diabetes, pregnancy-induced hypertension, and hyperemesis gravidaruim. By excluding these women, the study aimed to focus on a more homogeneous population, allowing for a clearer analysis of the relationship between GWG-related attitudes, pre-pregnancy weight and prenatal health behaviours.

## Sampling size and procedures

The required sample size was calculated using the following formula:

$$n = \frac{Z_{1-\alpha/2}2p(1-p)}{d^2}$$

where Z1-$\alpha$/2 is the standard normal variate for alpha 0.05 (1.96), P is the proportion of positive GWG attitude, and as its prevalence is unknown in Saudi Arabia, the (P) was considered (50%), and d is the precision (0.05). These parameters resulted in the minimum required sample size of 384. 15% was added to compensate for expected inconsistent data. The final responses were 413 participants (Fig. 1).

Participants were reached by disseminating the online questionnaire to pregnant women on social media groups (Facebook, Twitter, WhatsApp, Instagram, and Telegram specifically pregnancy concerned pages or groups). They were asked to fill out the questionnaire after obtaining informed consent and explaining the study's objectives and inclusion and exclusion criteria at the beginning of the online questionnaire. Reading the inclusion criteria and agreeing to the informed consent was mandatory before proceeding with the questionnaire and the principle investigator contact data was available for answering any questions. In addition to automated screening, the online survey was designed with built-in logic to filter out participants who did not meet inclusion criteria automatically. For example, if a participant answered "yes" to a question about having pregnancy induced hypertension or gestational diabetes, they were excluded from the study and did not complete the survey. Data were collected over 10 weeks from March to May 2024.

## Study instruments

The study's instrument was an online self-reported questionnaire developed after reviewing the relevant literature. The questionnaire includes three main parts:

Part I: participants' basic data includes demographic characteristics and reproductive history. It contains the woman's age, weight, and height before pregnancy, current weight, occupation, education, residence, and monthly income. Obstetric history includes gravity, parity, gestational age, antenatal care (ANC) regularity, and pregnancy planning.

Part II: Pregnancy and Weight Gain Attitude Scale (PWAS)

This scale was first developed by *Palmer, Jennings & Massey (1985)* to evaluate pregnant women's attitudes regarding pregnancy-related weight gain in North American pregnant women. The original scale contains 18 statements; in the current study, we used the French

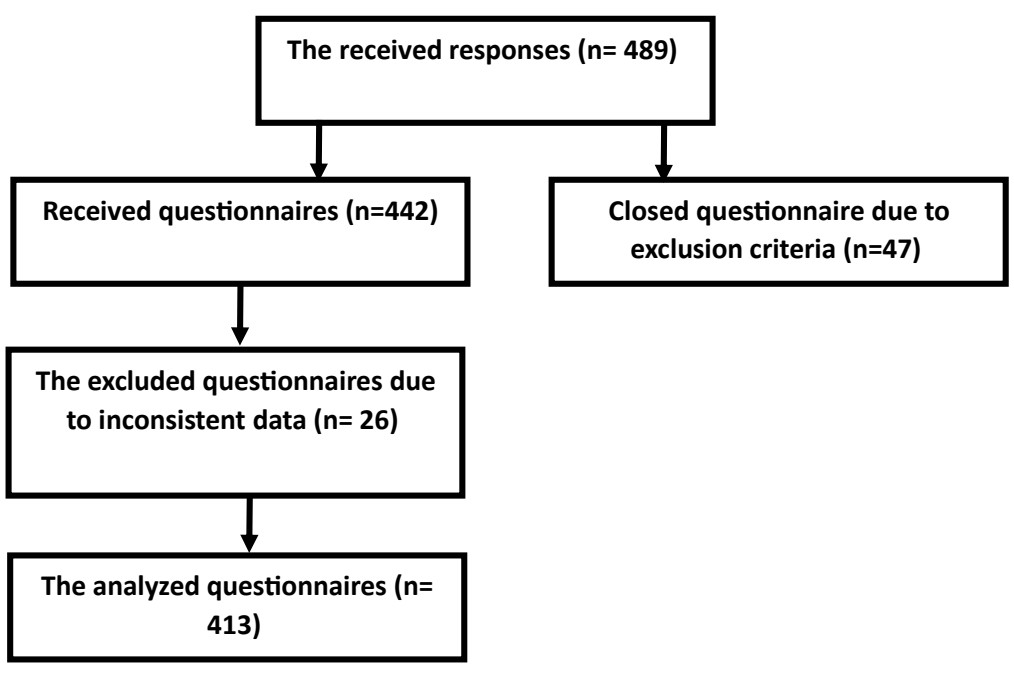

**Figure 1** Participants flow chart.

version validated by *Rousseau et al. (2016)*, which was translated into Arabic by an expert translator. The French version proposes a five-factor structure including 16 statements: fear about weight gain (four statements), absence of weight gain preoccupation (two statements), positive attitudes about weight gain (four statements), concern over weight gain (three statements), control over weight gain (three statements). The scale statements are rated on a 5-point Likert scale from 1 = strongly disagree to 5 = strongly agree. The overall score is computed by reversing eight negative items (1, 3, 9, 12–16). The mean score for each factor was calculated based on the numbers of statement. The overall score ranged from 16 to 80, with the optimal attitudes toward excessive pregnancy-related weight gain considered when the woman obtained three or less mean scores. A negative attitude toward EGWG is considered a motivator for pregnant women to adhere to a healthy lifestyle. *Rousseau et al. (2016)* evaluated the psychometric properties of PWAS and indicated satisfactory psychometric properties for the French version of the scale. After translation to Arabic, the PWAS was tested for validity by a jury of five experts in the maternity nursing field, as well as for reliability by the Cronbach's alpha coefficient test ($r = 0.877$).

Part III: Prenatal Health Behavior Scale (PHBS)

The 20-item PHBS is a self-report measure first developed by *DeLuca & Lobel (1995)* and used in numerous types of research (*Auerbach et al., 2017*; *Smith et al., 2020*; *Hill et al., 2019*) to estimate health-relevant behaviours during pregnancy, including smoking, diet, vitamin use, sleep, and physical activities. This scale consists of two main subscales: health-promoting behaviours (10 items) and health-impairing behaviours (10 items). In

PHBS, the pregnant woman was asked, "In the past two weeks, how often have you" engaged in each of the 20 behaviours with responses on a 5-point Likert scale, from 0 (Never) to 4 (Very Often). The total score for each subscale ranged from 0 to 40, with higher scores desired for the health-promoting behaviours subscale and lower scores desired for the health-impairing behaviours subscale. The reliability, validity, and suitability of the PHBS as a self-reported instrument for assessing health behaviours among pregnant women have been confirmed by *Auerbach et al. (2017)*.

### Ethical considerations

After the Deanship of Scientific Research approved the proposal, the proposal and data collection tools were sent to the main ethical committee at Najran University before starting data collection to obtain ethical approval (IRB registration number 202403-076-018912-042978). The study's objectives were written on the informed consent page at the front of the questionnaire, and the woman's agreement to informed consent was considered mandatory for answering the questionnaire. Anonymity was applied, and data confidentiality was maintained.

### Data analysis

The research data were analyzed using the SPSS version 23. Firstly, the responses were carefully checked for inconsistent data (such as data type mismatching values inconsistence, duplicated data and, data integrity issues) and normal distribution. Responses with inconsistent data were excluded from statistical analysis (17 responses excluded). The respondents' basic data, attitudes toward weight gain, and prenatal health behaviour were described using frequency and percentages for categorical variables and means and standard deviations (mean $\pm$ SD) for continuous variables. The overall scores of attitudes toward weight gain and prenatal health behaviours were obtained by summing items. One-way analysis of variance (ANOVA) was used to compare the attitudes toward GWG among pre-pregnancy BMI categories. Univariate and multivariate analyses examined the associations between GWG-related attitudes, pre-pregnancy BMI, and prenatal health behaviours. All variables with a $P$ value $\leq 0.25$ in the univariate analysis were involved in the multivariate model to control with all potential confounders. The model goodness of fit was examined using the Hosmer-Lemeshow and omnibus tests. The direction and strength of statistical relationships were assessed by the adjusted odds ratio (AOR) with 95% confidence interval (CI). The AOR and 95% CI were estimated to examine the associations between GWG attitudes and pre-pregnancy BMI and prenatal health behaviours in the binary logistic regression. The significance level was $p < 0.05$.

## RESULTS

### Participants' demographic characteristics and obstetrical history

Table 1 illustrates that the mean age of the study participants was 26.35 years, and approximately one-third (36.3%) of them had normal BMI before pregnancy, while the mean of their current pregnancy weight was 69.62 kg. A large proportion of the current study participants were urban area residents (90.7%), university-educated (81.4%), and

**Table 1   Participants' demographic characteristics and reproductive history (*n* = 413).**

| Demographic characteristics and reproductive history | f (413) | % |
|---|---|---|
| **Age in years (Mean ± SD)** | 26.35 ± 5.62 | |
| **Pre-pregnancy BMI** | | |
| Underweight (BMI < 18.5) | 62 | 15.0 |
| Normal weight (BMI = 18.5 = 25) | 150 | 36.3 |
| Overweight (BMI = 25. < 30) | 127 | 30.8 |
| Obese (BMI = 30 and more) | 74 | 17.9 |
| **Mean ±SD** | 25.69 ± 4.06 | |
| **Residence** | | |
| Urban | 375 | 90.7 |
| Rural | 38 | 9.3 |
| **Level of education** | | |
| Read and write | 12 | 2.9 |
| Secondary school | 65 | 15.7 |
| University or postgraduate | 336 | 81.4 |
| **Monthly income** | | |
| Not enough | 68 | 16.5 |
| Enough | 309 | 74.8 |
| Enough and save | 36 | 8.7 |
| **Working condition** | | |
| Working | 185 | 44.8 |
| Housewife | 228 | 55.2 |
| **Received education regarding GWG** | | |
| No | 324 | 78.5 |
| Yes, from health team | 42 | 10.2 |
| Yes, social media | 34 | 8.2 |
| Yes, Family and friends | 13 | 3.1 |
| **Gravidity (Mean ± SD)** | 1.79 ± 1.44 | |
| **Parity (Mean ± SD)** | 1.36 ± 1.24 | |
| **Gestational age in weeks (Mean ± SD)** | 31 ± 8.27 | |
| **Planned pregnancy** | | |
| Yes | 250 | 60.5 |
| No | 163 | 39.5 |
| **Regular ANC** | | |
| Yes | 343 | 83.1 |
| No | 70 | 16.9 |

had enough monthly income (74.8%). Furthermore, 21.5% of them received education regarding GWG, and more than half (55.2%) of them were housewives. Regarding reproductive characteristics, the mean gravidity, parity, and gestational age were 1.79 times, 1.36 times, and 31 weeks, respectively. In addition, 60.5% had planned for their pregnancy, and 83.1% had regular ANC.

### Attitudes toward GWG according to pre-pregnancy BMI categories

One-way ANOVA showed a significant correlation ($p < 0.05$) between pre-pregnancy BMI and fear of weight gain, positive attitudes toward EGWG, concern over weight gain, control over weight gain, and total GWG attitude. In addition, a significant relationship ($p < 0.05$) was found between pre-pregnancy BMI and practising health-promoting behaviours during pregnancy, as shown in Table 2.

### Logistic regression analysis for positive GWG attitudes

Logistic regression analysis showed that abnormality in pre-pregnancy BMI negatively predicts a positive attitude toward EGWG. In detail, being overweight [AOR = 0.434, $P = 0.017$, 95% CI [0.219–0.861]] or obese [AOR = 0.157, $P = 0.001$, 95% CI [0.052–0.469]] significantly decreased the probability of positive attitude toward EGWG when compared with normal weight woman before pregnancy. Furthermore, practising health-promoting behaviours during pregnancy [AOR = 0.132, $P = 0.001$, 95% CI [0.050–0.483]] significantly decreased the probability of a positive attitude toward EGWG. Lastly, women who received information regarding optimal weight gain during pregnancy have a lower probability of having a positive attitude toward EGWG than women who did not receive information [AOR = 0.201, $P = 0.001$, 95% CI [0.163–0.861]]. Regarding demographic factors, age and educational level are the main predictors that affect women's perception of positive GWG attitudes. An increment in a woman's age by one year increases the woman's tendency to have a positive attitude toward EGWG by 2.9 times. In addition, women with a higher education have a lower tendency to have a positive attitude toward EGWG ($p < 0.05$). Concerning obstetric history, increased gravidity and parity can significantly decrease the women's probability of a positive attitude toward EGWG ($p < 0.05$). Furthermore, women who attend regular antenatal care have a lower tendency toward a positive attitude toward EGWG compared to women who did not attend regular antenatal care ($p < 0.05$), as illustrated in Table 3.

## DISCUSSION

The relationship between below and over the recommended GWG and pregnancy outcome is complex and multifactorial and mostly linked to pregnancy complications, either physical or psychological. For example, women with pre-pregnancy obesity with high GWG are at higher risk for numerous pregnancy-related complications, including gestational diabetes, pregnancy-induced hypertension, premature labour, postpartum depression, and other dangerous outcomes compared with women with normal pre-pregnancy weight who gained moderate gestational weight (*Santos et al., 2019*).

The current study investigated the link between pre-pregnancy weight, practising healthy behaviours during pregnancy, and the attitude toward GWG. One-way ANOVA showed a significant correlation between pre-pregnancy BMI, fear of weight gain, positive attitude toward EGWG, concern over weight gain, control over weight gain, and total GWG attitude. In addition, a significant relationship was found between pre-pregnancy BMI and practising health-promoting behaviours during pregnancy.

**Table 2   Attitudes toward excessive GWG and PHB according to pre-pregnancy BMI categories.**

| | Pre-pregnancy BMI categories (kg/m$^2$) | | | | | |
| | Underweight (BMI < 18.5) | Normal weight (BMI = 18.5 = 25) | Overweight (BMI = 25. < 30) | Obese (BMI = 30 and more) | F | *p* |
|---|---|---|---|---|---|---|
| **Pregnancy and weight gain attitude scale (PWAS)** | | | | | | |
| Factor 1 Fear of weight gain | 2.04 ± 0.83 | 2.62 ± 0.93 | 2.95 ± 1.06 | 2.36 ± 0.79 | 11.200 | 0.000[*] |
| Factor 2: Absence of weight gain preoccupation | 3.98 ± 0.85 | 3.96 ± 0.71 | 3.96 ± 0.80 | 4.20 ± 0.36 | 0.647 | 0.585 |
| Factor 3: Positive attitudes toward excessive weight gain | 4.00 ± 0.86 | 3.30 ± 0.98 | 3.08 ± 1.02 | 3.25 ± 0.48 | 11.69 | 0.000[*] |
| Factor 4: Concern over weight gain | 2.01 ± 0.77 | 2.23 ± 0.89 | 2.51 ± 1.15 | 2.20 ± 0.74 | 3.474 | 0.016[*] |
| Factor 5: Control over weight gain | 3.09 ± 0.37 | 3.23 ± 0.62 | 3.46 ± 0.63 | 3.15 ± 0.91 | 4.615 | 0.003[*] |
| Total GWG attitude | 3.02 ± 0.14 | 3.07 ± 0.32 | 3.19 ± 0.40 | 3.03 ± 0.41 | 3.868 | 0.010[*] |
| **Prenatal Health Behavior Scale (PHBS)** | | | | | | |
| Total health-promoting behaviors | 2.52 ± 1.22 | 3.17 ± 1.08 | 3.14 ± 1.10 | 2.64 ± 0.81 | 6.952 | 0.000[*] |
| Total health-impairing behaviors | 3.79 ± 1.02 | 3.68 ± 0.84 | 3.61 ± 1.00 | 3.72 ± 0.54 | 0.460 | 0.710 |

**Notes.**
F, One-way ANOVA.
*significant at *p* = 0.05.

**Table 3  Logistic regression analysis for positive attitude toward EGWG.**

| Variable | Univariate | | | Multivariate | | |
|---|---|---|---|---|---|---|
| | AOR | P value | 95% CI | AOR | P value | 95% CI |
| **Pre-pregnancy BMI** | | 0.010[*] | | | 0.009[*] | |
| Normal weight | Ref | | | Ref | | |
| Underweight (BMI < 18.5) (BMI=18.5 = 25) | 0.501 | 0.087 | 0.238–1.100 | 0.486 | 0.069 | 0.220–1.076 |
| Overweight (BMI = 25 < 30) | 0.445 | 0.020[*] | 0.225-0.879 | 0.434 | 0.017[*] | 0.219-0.861 |
| Obese (BMI = 30 and more) | 0.160 | 0.001[*] | 0.054–0.478 | 0.157 | 0.001[*] | 0.052–0.469 |
| **Prenatal Health Behavior** | | | | | | |
| Total health-promoting behaviors | 0.158 | 0.001[*] | 0.053–0.477 | 0.132 | 0.001[*] | 0.050–0.483 |
| Total health-impairing behaviors | 0.863 | 0.320 | 0.682–1.092 | | | |
| **Received education regarding GWG** | | | | | | |
| No | Ref | | | | | |
| Yes | 0.230 | 0.001[*] | 0.152–0.873 | 0.201 | 0.001[*] | 0.163–0.861 |
| **Age in years** (Mean ±SD) | 2.949 | 0.004[*] | 1.915–3.237 | 2.932 | 0.004[*] | 1.887–3.127 |
| **Residence** | | | | | | |
| Urban | Ref | | | | | |
| Rural | 0.498 | 0.397 | 0.099–2.501 | | | |
| **Level of education** | | | | | | |
| Read and write | Ref | | | | | |
| Secondary school | 0.432 | 0.030[*] | 0.231–0.881 | 0.435 | 0.019[*] | 0.221–0.872 |
| University or postgraduate | 0.171 | 0.002[*] | 0.130–0.494 | 0.145 | 0.001[*] | 0.152–0.478 |
| **Monthly income** | | 0.505 | | | | |
| Not enough | Ref | | | | | |
| Enough | 0.903 | 0.720 | 0.516–1.580 | | | |
| Enough and save | 1.435 | 0.437 | 0.578–3.563 | | | |
| **Working condition** | | | | | | |
| Working | Ref | | | | | |
| Housewife | 0.832 | 0.380 | 0.552–1.254 | | | |
| **Gravidity** | 0.808 | 0.004[*] | 0.699–0.934 | 0.817 | 0.041[*] | 0.687–0.923 |
| **Parity** (Mean ±SD) | 0.735 | 0.000[*] | 0.623–0.868 | 0.726 | 0.014[*] | 0.562–0.938 |
| **Gestational age in weeks** (Mean ±SD) | 1.036 | 0.426 | 0.950–1.128 | | | |
| **Planned pregnancy** | | | | | | |
| Yes | Ref | | | | | |
| No | 1.069 | 0.755 | 0.703–1.626 | | | |
| **Regular ANC** | | | | | | |
| No | Ref | | | | | |
| Yes | 0.416 | 0.027[*] | 0.191–0.906 | 0.418 | 0.029[*] | 0.213–0.917 |

**Notes.**
AOR, Adjusted Odd Ratio; CI, Confidence Interval.
[*]significant at $p = 0.05$.

The relationship between pre-pregnancy BMI and GWG attitude is complex and can result in body dissatisfaction and poor body image (*Andrews, Hill & Skouteris, 2018*). The link between pre-pregnancy BMI and health-promoting behaviours indicates the impact of a healthy lifestyle followed before and continued during pregnancy. In addition, numerous

women do not give too much importance to weight control during pregnancy as they report other priorities at this time. These priorities include maintaining a healthy baby and complication-free, comfortable pregnancy (*Vanstone et al., 2020*). Even if unintended, such priorities can help women practice health-promoting behaviours and maintain a healthy weight. However, women's priorities during pregnancy will change according to pregnancy-related circumstances rather than being planned. Furthermore, the psychological control of weight gain attitude is very important and can help to shape the woman's health behaviours to control weight. Risk factors that enhance EGWG include negative body image and a higher target of weight gain during pregnancy. Many women hold the misconception that pregnant women should double their food intake. Furthermore, inaccurate perceptions exist regarding recommended GWG and perceived obstacles to maintaining a balanced diet. Conversely, factors that may help prevent EGWG include strong sense of personal control over weight gain and adherence to recommended GWG (*Kapadia et al., 2015*).

Logistic regression analysis showed that being overweight or obese significantly decreased the probability of a positive attitude toward EGWG when compared with normal weight woman before pregnancy. The relationship between pre-pregnancy BMI and GWG attitude is rarely investigated. However, Andrews et al. concluded a significant mediating relationship between pre-pregnancy BMI and GWG and body attitude. Therefore, they emphasized the role that prenatal care can play in promoting a positive body attitude and appropriate GWG, especially for women entering pregnancy with BMI outside the healthy range. In detail, obese women entered pregnancy with a negative GWG attitude, while normal weight women started pregnancy with a positive attitude toward EGWG. In addition, increased stress around GWG, increased body dissatisfaction, and negative attitudes toward GWG were associated with pre-pregnancy BMI abnormalities (*Andrews, Hill & Skouteris, 2018*). Furthermore, a Japanese study found that pregnant women in their study were classified into three groups in relation to their GWG. The first group includes older multiparous women who entered pregnancy and had a history of pre-pregnancy underweight; therefore, they are concerned only with safe delivery and a healthy newborn and pay no or little concern to GWG. The second group consists of younger, multiparous women who practice healthy lifestyles and have normal BMI before pregnancy, so they continue to be concerned about body shape, healthy lifestyle, GWG, delivery, and newborn health. The third group is multiparous less educated smokers who have unhealthy lifestyles and abnormal pre-pregnancy BMI, and they will continue the unhealthy lifestyle during pregnancy and consequently have no concern regarding GWG. They further added that low pre-pregnancy BMI did not add more risk to newborn weight. On the other hand, according to *NA et al. (2021)* women who did not adhere to GWG guidelines gained more weight overall but gave birth to babies with lower birth weights compared to those who followed the recommendations. The pre-pregnancy weight and healthy lifestyle practised before pregnancy seem to be the most important motivators to the woman to continue during pregnancy, as the woman will not start new habits during pregnancy except if she fears pregnancy-related complications. Another study found that although healthcare providers told pregnant women about the recommended GWG and EGWG complications, pre-pregnancy obesity is still a predictor of EGWG (*Lott et al., 2019*). A recent study

confirmed a strong relationship between pre-pregnancy BMI and GWG attitude, as obese women had lower GWG scores when taking normal weight as a reference, indicating that obese women had a negative attitude toward GWG (*Savard et al., 2021*).

Furthermore, practising health-promoting behaviours during pregnancy significantly decreases the probability of a positive attitude toward EGWG. However, a previous study reported that all women were concerned about healthy lifestyles and GWG during pregnancy, which continued from their pre-pregnancy lifestyle, and no woman started to be concerned about weight gain during pregnancy (*Vanstone et al., 2017*; *Vanstone et al., 2020*). In addition, a Pennsylvanian study found that pregnant women's counselling regarding the importance of weight control motivates them to practice healthy behaviours and have a negative attitude toward EGWG during pregnancy (*Lott et al., 2019*). Therefore, healthcare providers can play a crucial role in improving pregnant women's health behaviours and improving pregnant women's weight control attitude.

Moreover, women who received information regarding normal GWG are less likely to have a positive attitude toward EGWG than women who did not receive information; only 21.5% of our participants received education regarding GWG. Healthcare providers are responsible for informing pregnant women about the recommended GWG and the complications of being overweight. In addition, women's pre-pregnancy concerns regarding weight gain can extend to their healthy behaviours during pregnancy (*Vanstone et al., 2020*). In a systemic review conducted by *Vanstone et al. (2017)*, the women's main priority during pregnancy is to maintain the fetus's well-being, and they are ready to do anything to achieve this priority. Unfortunately, most women did not know the well-established link between EGWG and all types of pregnancy-related complications and negative fetal outcomes. If such a link is elaborated, women will be ready to do anything to reduce weight gain above the permissible limit during pregnancy. Furthermore, informing women of this important information from the first day of pregnancy can promote positive health behaviours and modify their attitude, thus GWG (*Vanstone et al., 2017*). Furthermore, most healthcare providers provide GWG-related health education only to women who are at high risk, obese, or underweight. Many hidden weight-related risk factors may be associated with pregnancy; therefore, education regarding GWG should be provided as an essential part of antenatal education for all pregnant women (*NA et al., 2021*).

On the contrary, *Lott et al. (2019)* studied the GWG attitude and physical exercises among pregnant women in Pennsylvania. They reported that 78.8% of the participants received health education about diet and weight gain from healthcare providers. They added that 81.6% had health education regarding the recommended weight gain during pregnancy. In addition, the health care providers told them about complications from EGWG for the fetus, mother, and the progress of labour (*Lott et al., 2019*). The difference between the current study findings and the latter may be related to cultural differences between Saudi Arabia and Pennsylvania. In Saudi Arabia, increased pregnant woman weight is considered a good sign of a healthy woman and baby. The antenatal care provided to pregnant women concentrates on physical examination and lab investigations and pays little attention to health education. Health education is an important component of antenatal

care and needs more attention from healthcare providers (*Dayyani, Lou & Jepsen, 2022*; *Sparks et al., 2024*).

Although research has not addressed the relationship between EGWG attitudes and sociodemographic characteristics, our study confirmed associations between sociodemographic variables (such as age and education level) and GWG attitudes, suggesting that highly educated women have a negative attitude toward EGWG. Indeed, highly educated women may experience increased anxiety about EGWG due to their greater access to knowledge and increased awareness of the risks associated with EGWG. However, older women tend to have a more positive attitude toward weight gain during pregnancy, thanks to their accumulated life experiences, which enhance their understanding of health and bodily changes. This attitude allows them to prioritize their personal and physical well-being over societal pressures related to body image. The increased self-acceptance and positive body image that develop with age also reduce the anxiety and fear associated with weight gain and make them less concerned with beauty standards. Furthermore, the current findings indicate that increased pregnancies and births are associated with a lower likelihood of adopting positive attitudes toward EGWG, likely due to women's personal experience with weight control challenges and potential complications from previous pregnancies. This shift in attitudes is attributed to the increased frequency of consultations with healthcare providers and greater awareness of maternal and neonatal health threats. Similarly, women who regularly attend prenatal care are less likely to adopt positive attitudes toward EGWG; this may be due to easier access to healthcare, which facilitates education and follow-up and supports healthy weight control throughout pregnancy.

## Strengths and limitations

The current study has several strengths. First, it clarified important associations that are rarely studied, especially in Saudi Arabia, using all components of prenatal health behaviours. Second, univariate and multivariate logistic regression analyses were estimated to explore the association between GWG attitudes, pre-pregnancy BMI, and prenatal health behaviours. However, the self-reported nature of the online questionnaire may introduce recall bias. In addition, convenience sampling is subject to selection bias, which negatively affects the generalizability of the results. Furthermore, disseminating the questionnaire *via* social media platforms limits its accessibility to all pregnant women.

## CONCLUSIONS

This study showed that the pre-pregnancy BMI was significantly associated with GWG attitude. More than one-third of the participant had unplanned pregnancies. Elevated pre-pregnancy BMI is negatively predicts a positive attitude toward EGWG. Furthermore, women who received information about recommended GWG and women who practised health-promoting behaviours during pregnancy significantly reduced the likelihood of having a positive attitude toward EGWG. Therefore, to maintain an appropriate GWG attitude during pregnancy, healthcare providers must be proactive and start early by securing a normal BMI before pregnancy and educating women about systematic health promoting behaviours, not just prenatal, would have a stronger impact.

## ACKNOWLEDGEMENTS

The research team expresses gratitude to the study participants for their voluntary participation.

### Funding

The Deanship of Graduate Studies and Scientific Research at Najran University, grant code (NU/EP/MRC/13/23). The funders had no role in study design, data collection and analysis, decision to publish, or preparation of the manuscript.

### Grant Disclosures

The following grant information was disclosed by the authors:
Deanship of Graduate Studies and Scientific Research at Najran University: NU/EP/MRC/13/23.

### Competing Interests

The authors declare there are no competing interests.

### Author Contributions

- Wafaa Taha Elgzar conceived and designed the experiments, performed the experiments, prepared figures and/or tables, authored or reviewed drafts of the article, and approved the final draft.
- Majed Said Alshahrani performed the experiments, prepared figures and/or tables, authored or reviewed drafts of the article, and approved the final draft.
- Nouf Salem Alshahrani analyzed the data, prepared figures and/or tables, authored or reviewed drafts of the article, and approved the final draft.
- Heba Abdel-Fatah Ibrahim conceived and designed the experiments, performed the experiments, analyzed the data, prepared figures and/or tables, authored or reviewed drafts of the article, and approved the final draft.

### Human Ethics

The following information was supplied relating to ethical approvals (i.e., approving body and any reference numbers):

The study was approved by the main ethical committee at Najran University.

### Data Availability

SPSS program.

### Supplemental Information

Supplemental information for this article can be found online at http://dx.doi.org/10.7717/peerj.19349#supplemental-information.

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
