# Peer review of "Association between gestational weight gain-related attitudes, pre-pregnancy weight and prenatal health behaviours"

_PeerJ, doi:10.7717/peerj.19349_

## Round 0.1 · original submission · Major Revisions

Dear Dr. Ibrahim,

Your manuscript entitled “Association between gestational weight gain-related attitudes, pre-pregnancy weight and prenatal health behaviours", which you submitted to PeerJ, has been reviewed by the editor and 3 external reviewers.

The reviewers have raised significant concerns that must be addressed before the manuscript can be considered further. However, since two reviewers found some merit in the paper, and the most critical reviewer noted that this “is novel research and there is an opportunity for it to be a contribution to the field,” I am willing to reconsider if you undertake major revisions and resubmit.

If you decide to resubmit the revised version, please summarize all the improvements made in the new version and give answers to all critical points raised in the reviewers’ report in an accompanying letter. Copy and paste each and every reviewer's comment above your response. A native English speaker should also check the manuscript for spelling and grammar.

Please note that resubmitting your manuscript does not guarantee eventual acceptance. Since the requested changes are major, the revised manuscript will undergo a second round of review by the same reviewers. I must emphasize that the acceptability of the revision will depend upon the resolution of the points raised by the reviewers.

Sincerely yours,

Stefano Menini

·

Basic reporting

English Language: The manuscript is generally written in professional English, but there are areas where clarity and fluency can be improved. For instance, simplifying the phrasing in the introduction (lines 16-20) would help. There are minor grammatical errors, but they do not obscure the meaning.
Literature Review: The literature review provides sufficient context but would benefit from the inclusion of more recent studies related to gestational weight gain (GWG) and its impact on pregnancy outcomes. I recommend the addition of the following two articles to enhance the relevance of the literature:
The impact of gestational weight gain on fetal and neonatal outcomes: the Araraquara Cohort Study. BMC Pregnancy and Childbirth. 2024; 24: 320. doi: 10.1186/s12884-024-06523-x.
Predictors of inadequate gestational weight gain according to IOM recommendations and INTERGROWTH-21st standards: the Araraquara Cohort Study. BMC Pregnancy and Childbirth. 2024; 24: 579. doi: 10.1186/s12884-024-06545-x.
Figures and Tables: The figures and tables are clear, well-labeled, and relevant to the study. However, please ensure that none of the data have been manipulated inappropriately. The values presented in Tables 1-3 should be cross-checked with the raw data to ensure accuracy.
Raw Data: The raw data file needs to be reviewed for accessibility and appropriate description. It should align with the results presented in the manuscript and be organized clearly for replication.

Experimental design

Research Question: The research question is well-defined, relevant, and addresses a significant knowledge gap, particularly in the context of Saudi Arabian women. The study fits within the journal's scope.
Ethical Standards: The study was conducted in line with ethical standards, with appropriate approval from the ethics committee (IRB registration number: 202403-076-018912-042978). Informed consent procedures are clearly described.
Methods: The methodology is sound, with sufficient detail provided for replication. However, I suggest elaborating on how covariates were chosen for the multivariate model, as this would improve the clarity and rigor of the methods. The authors should also discuss potential biases related to self-reported data.

Validity of the findings

Data Soundness: The findings are statistically sound and robust. The use of logistic regression is appropriate given the research question. The data support the conclusions drawn, and all necessary statistical tests appear to have been conducted.
Conclusions: The conclusions are clear and well-supported by the results. However, the discussion would benefit from a more detailed examination of the limitations of the study, including the use of self-reported data and the convenience sampling method. Addressing these limitations would enhance the transparency of the study.

Additional comments

Strengths: This study addresses an important public health issue, particularly in Saudi Arabia, where rates of obesity among pregnant women are high. The use of validated tools, such as the Pregnancy and Weight Gain Attitude Scale (PWAS) and the Prenatal Health Behavior Scale (PHBS), strengthens the study's validity.
Weaknesses: One potential weakness is the reliance on self-reported data, which can introduce bias. Additionally, the convenience sampling technique is not discussed in detail as a limitation.
Suggestions for Improvement:
The authors should expand on the cultural factors that may influence GWG attitudes in Saudi Arabia. This would provide a more nuanced interpretation of the results.
The rationale behind the exclusion criteria, particularly the exclusion of women with pregnancy complications such as gestational diabetes and hypertension, should be further clarified.
Consider discussing the limitations of the convenience sampling technique in greater detail.

Reviewer 2 ·

Basic reporting

- Please make sure you use more inclusive and less stigmatizing words around weight. For example, I would avoid using terms like “abnormality” line 28, “abnormal” line 228, and “appropriate” line 63.
- The authors did a good job about reporting on the prevalence of overweight and obesity in Saudi Arabia. Since the focus of the paper is on gestational weight gain, however, it will be helpful to include data on GWG within the Saudi context.
- I would recommend that the paper get reviewed or edited for English language by professional service.

Experimental design

- It is not clear how the sample size of 413 was achieved. In other words, how many women were approached? How many met the inclusion criteria? How many consented to the study? Was there any data missingness?
- It is also not clear how they screened women. There were specific inclusion and exclusion criteria but no information on how these criteria were assessed through the online questionnaire.
- It will be helpful to add more information about the online recruitment. How did they reach out to pregnant women specifically? Were there participation incentives?
- More information about the Pregnancy and Weight Gain Attitude Scale is needed. Based on my understanding, the scale had values range from 16 to 90 (16 questions with individual question scores ranging from 1 to 5). Is this correct? What did the authors mean by “For the overall scores, the optimal attitudes to pregnancy-related weight gain are considered when the woman obtained three or more mean scores”. It is also not clear whether higher scores or lower scores on the scale represent a better perception about GWG.
- Similarly, for the “Prenatal Health Behavior Scale (PHBS)” readers need to understand what does low and high scores mean.

Validity of the findings

- The phrase “positive attitudes toward excessive weight gain” has been repeatedly used in the paper. It will be helpful to explain what does it mean. Similarly in line 182/183 the authors mention that pre-pregnancy BMI “negatively predicts a positive attitude toward excessive GWG”. This is also confusing to reader.
- The authors should clearly state what are their outcomes and what does negative and positive deviations from these outcomes signify.
- In table 2, can you please explain what factors 1 to 5 meant and how the scoring for each was developed?
- In Table 3, it is not clear what variables were adjusted for. Did the authors control for age, residence, income, level of education, etc…?
- Can the authors provide explanation why underweight rather than normal weight was used as the reference category in table 3?
- Limitations of the study should be addressed in more depth.

Additional comments

The present paper investigates the association between pregnancy BMI and prenatal health behaviors on gestational weight gain attitudes. The study is unique in Saudi Arabia as it has not been previously investigated there. The study lacks however important information about the methodology section which makes it difficult to evaluate the results of the study. More information is needed on: i) the selection of the study participants, ii) definitions of the outcome variables and its measurement, and iii) statistical analysis.

Reviewer 3 ·

Basic reporting

Throughout the document, to be more gender inclusive, please adopt the language, ‘pregnant person’, throughout the manuscript

The discussion section is very lengthy and does not target specific aspects of the study. It needs to be rewritten and condensed to focus on the supportive evidence for the findings, the strengths and limitations, and future directions.

There are multiple typographical errors throughout the document that need to be addressed to improve readability and coherence of the manuscript. Generally speaking, when presenting the full name before indicating the abbreviation (e.g. gestational weight gain (GWG) antenatal care (line 133: ANC) you do not necessarily need to capitalize the full name: only if it is a proper noun. Additionally: line 81: GWG Attitudes (attitudes is not capitalized).

Specific line-item feedback:

Please review grammatical structure throughout the document. For example:
E,g, line 75: Although the appropriate GWG
Line 86: no period after trimester
Line 130: Participants' basic data
Line 186: one-third (36.3%) of them

Maternal overweight is not a correct syntax: mothers who are overweight

Line 66: “A pregnant woman must adapt to the major change in her body image as a result of GWG.” This is a subjective statement and has considerable implications. Please rework/remove and support statements with potential subjective implications with scientific evidence.

Line 153 “used in numerous types of research”: please be more clear

Line 177: Please correct language. One does not deal with a confounder, but address or control for them

Line 188: “enough monthly income”: this is not appropriate language for reporting income, nor does it provide sufficient detail to quantify what is ‘enough’. Please provide further detail to quantify this metric.

Line 198: Table 2: please be clear which data is being presented in what table (PWAS or PHBS).

Line 208: Preference to use the term optimal versus normal weight gain

Line 228-230: this is a subjective statement not supported by evidence. is also contradictory to the following statement in the paragraph. This needs to be reconsidered/removed.

Line 237-242: This paragraph lacks clarity. Please review or remove.

Line 263: How can one be certain it was ignored? Please clarify.

Line 263-264: Please use appropriate terminology: not ‘lighter’ babies. Does this mean low birth weight?

Line 322: When stating abnormal pre-pregnancy BMI, does it imply elevated? Since the study does not address low BMI, indicating abnormal pre pregnancy BMI is misleading

Line 331: Is the project code missing?

Experimental design

Line 45: please define what is considered excessive gestational weight gain in this sentence.

Line 57: The authors define according to IOM standards what optimums GWG is , but there should also be an indication of what the categories are for pre-pregnancy BMI, and how it is calculated, since it is fundamental to the study. Also, the values listed are the total gestational weight gain in pregnancy. Since the study included women in both their second and third trimester, it should look at the weight gain by trimester and not total GWG. If this correction factor was not implemented in the design, this is a fundamental issue with the design.

Line 59: Is the cohort study reported by Al-Wahabi et al representative of the entire population of Saudi Arabia? This is important given the cohort is from this population, and if this data is being used to support the rationale, please indicate indicate this.

Line 64-65: “rapid economic development and expansion of urbanization, which has led to a lifestyle that depends on a diet rich in carbohydrates and fats and a lack of physical fitness”: what is the impact of socioeconomic status on this statement? Given the association to maternal weight, it is important to address.

Line 86-89: I don’t feel that the study has presented the rationale sufficiently to support this statement. Elevated GWG is not a body image issue but a physical status, and while the associated with high pre-pregnancy BMI and adverse health outcomes is noted, rationale is not sufficiently presented to this point.

Line 92-93: The objective of this study is not well presented nor is the rationale provided to support this study. It is confusing to ascertain the study addresses maternal weight, attitudes towards weight gain, self image (which has other psychological factors such as confidence, anxiety, depression, etc.). Please very clearly articulate the objectives, with the evidence to support it.

Line 94-96: . “If statistical data confirms such a relationship, it will direct healthcare providers toward proactive strategies and start early by securing a normal BMI before pregnancy and educating women about health-promoting behaviours”. This is overstating the impact of the findings. There is implication of causation which the analysis cannot support. Also, is the study addressing pre pregnancy BMI or gestational weight gain? Again, it is very confusing, and this directive suggest both.

Line 100: Weight and BMI are being used interchangeably when they are not interchangeable (body weight is using in the calculation of BMI).

Line 124: Was there compensation for participating?

Line 166: Please provide further details on how anonymity was applied. Was it truly anonymous, or deidentified?

Line 169: Please elaborate. What criteria was applied to be considered inconsistent data? Also, was a normal distribution expected for all responses? There needs to be additional justification for excluding data.

Line 171: Please indicate what is meant by ‘clarified’ in the data analysis: this is not well understood.

Line 174: Please provide justification for using a one way and not a two-way ANOVA.

Line 175: Please be more specific when outlining which univariate and multivariate analyses were used in the analysis. Much more detail is required. Also, it is univariate, not univariant analysis. Please correct throughout manuscript.

Line 180: The unadjusted odds ratio needs to be completed before completing an adjusted odds ratio. Also, confounders are not clearly listed in the AOR model.

Line 187: I fail to see why current weight is included. The study is addressing BMI/GWG therefore should report these weight related variables. This weight metric does not have context re trimester, height etc. Also, kilogram should be reported as kg not Kg.

The manuscript indicates that that variables needed to reach p <0.25 for inclusion (line 177), but used 0.05 in the one way ANOVA (line 194). Please be clear why these two values were chosen.

Line 210: “women who did not receive information”: As this is an important factor in the analysis, there needs to be more detail of where they received information (medical professional, online, etc) and what information was received

Validity of the findings

It is stated in line 17 and again in line 214 that gestational weight gain and pregnancy outcomes is …”mostly linked to pregnancy complications..”. Please provide justification, because as stated it is not correct. There may be an associated relationship between above and below recommended GWG and adverse pregnancy outcomes, but GWG in itself is not associated with complications.

Line 107: please address the numerous biases associated with convenience sampling technique, as they have a direct impact on the interpretability of the results.

Line 109-112: Please indicate why the study only included women in their second and third trimesters

Line 109-112: Women were excluded who have comorbidities associated with elevated GWG and BMI, thereby biasing the sample. The study could have corrected for these factors within the statistical analysis.

Line 121-122: Disseminating the questionnaire via social media platforms is immediately biasing the sample, as not all pregnant women may have access or proficiently in navigating this platform. Please see previous comment on bias associated with convenience sampling.

Line 122-124: Please be specific how informed consent was obtained. If it was strictly done via an online consent form, how can it be certain it was informed if the participant did not have an opportunity to ask questions?

Line 126: Understand the limited time frame given the study design, but need to consider the limitations associated with such a restricted time frame, including time period bias.

Line 131: When considering current weight, was a correction factor applied for the participant’s trimester of pregnancy? Please provide rationale if not.

Line 132-133: Please confirm why each aspect of obstetric history was necessary for this study (e.g. pregnancy planning). It is not evident from the manuscript.

Line 130-133: It is important to address issues associated with self report and recall bias with this data collection for Part 1.

Line 140: Please include rationale of why the French version was translated. Also, do the items differ from the English version? If not, no need to indicate that the “french version proposes…”

Line 150: The study includes the psychometric properties of the Spanish version of the PWAS, but has the Arabic version had the psychometric properties validated? A direct translation does not ensure that the psychometric properties are maintained. If not, this could be a fundamental flaw in the research design as the study cannot attest to the reliability, validity or sustainability of the tool used in the study. There is literature available on how to ensure psychometric properties are maintained when instruments are translated (e.g. Cruchinho et al 2024, Kristjansson et al 2003, Hilton et al 2002).

Line 159: Please indicate the psychometric properties of the specific version used in this research (e.g. if Arabic version was used, please provide psychometric properties for this version). To say it is confirmed does not indicate the appropriateness for it’s use.

Line 315 states that the standardized data measurement instruments is a strength of the study, but unless they are standardized in the language of administration, this cannot be stated.

Table 2: The results suggest an overall association, but not by BMI group for the PWAS. Suggest sub analyses to ascertain exact relationship here. Additionally, it is indicated in line 219 that there is a link to pp BMI, but there needs to be a sub analysis to present this as a finding.

Table 3: It is a fundamental flaw to have underweight as the reference category in this analysis, as underweight could also present with issues concerning the health behaviours. Please provide rationale, but recommend using the normal weight as the reference category.

Line 221: The findings of this study, supported by an ANOVA study, is not sufficient to state a significant correlation and strong support for the findings (particularly when considering above noted concerns in the design and methodology). Consider more extensive epidemiological, correlational analysis techniques to strengthen this argument.

Line 318: There are numerous biases in this study that need to be further explored. If the data is truly anonymous, social desirability would be minimal.

Line 319: The study design has inherent limitations that must be outlined here, as well as the sampling framework, the exclusion criteria, and the analyses.

Line 326-327: The conclusions need further consideration. 1. a high proportion of pregnancies are unplanned; 2. it may take time to reach a normal BMI, which may not be available if a person is trying to get pregnant. 3. Systematic health promoting behaviours, not just prenatal, would have a stronger impact.

Additional comments

Thank you for the opportunity to review this manuscript. It is novel research and there is an opportunity for it to be a contribution to the field. However, there are major aspects of this study that must be addressed before it is appropriate for publication. The objective of the study is not clearly presented, and there are limitations in the design sampling framework, and exclusion criteria which introduces bias that needs to be addressed to appropriately interpret the results. The instruments used for the data collection do not included psychometric properties for the language version in which they were administered, therefore it is unknown if the instruments were in fact psychometrically sound. Further, there are limitations to the analyses which impacts the findings. For example, the authors apply overall GWG but included pregnant persons in both the second and third trimester. Also, it is not clear why certain data was excluded from the analyses. Additionally, underweight was listed as the reference category in the statistical model, but this group also has an associated elevated risk. In terms of presentation, there are numerous typographical and grammatical issues throughout the manuscript. Also, the discussion section should be revisited as it is lengthy and unfocused.
While I believe this manuscript presents an opportunity to advance the scientific evidence in this field, the study needs a significant review in light of the limitations and feedback noted above. Once addressed it may be appropriate for resubmission.

---

## Round 0.2 · accepted · Accept

Dear Dr. Ibrahim,

Thank you for submitting the revised version of your manuscript. I have personally reviewed the revision and confirmed that all the reviewers' comments have been adequately addressed. The quality of the manuscript has significantly improved as a result. I am pleased to inform you that your manuscript is now ready for publication in PeerJ in its current form.

Sincerely yours,

Stefano Menini